# Osmotic Cloud-Edge Intelligence for IoT-Based Cyber-Physical Systems

**DOI:** 10.3390/s22062166

**Published:** 2022-03-10

**Authors:** Giuseppe Loseto, Floriano Scioscia, Michele Ruta, Filippo Gramegna, Saverio Ieva, Corrado Fasciano, Ivano Bilenchi, Davide Loconte

**Affiliations:** 1Department of Management, Finance and Technology, LUM University “Giuseppe Degennaro”, Strada Statale 100 km 18, I-70010 Casamassima, Italy; loseto@lum.it; 2Department of Electrical and Information Engineering, Polytechnic University of Bari, Via E. Orabona 4, I-70125 Bari, Italy; floriano.scioscia@poliba.it (F.S.); filippo.gramegna@poliba.it (F.G.); saverio.ieva@poliba.it (S.I.); corrado.fasciano@exprivia.com (C.F.); ivano.bilenchi@poliba.it (I.B.); davide.loconte@poliba.it (D.L.); 3Exprivia S.p.A., Via A. Olivetti 11, I-70056 Molfetta, Italy

**Keywords:** Cloud-Edge Intelligence, Edge AI, microservice architecture, Osmotic Computing, Cyber-Physical Systems, Internet of Things

## Abstract

Artificial Intelligence (AI) in Cyber-Physical Systems allows machine learning inference on acquired data with ever greater accuracy, thanks to models trained with massive amounts of information generated by Internet of Things devices. Edge Intelligence is increasingly adopted to execute inference on data at the border of local networks, exploiting models trained in the Cloud. However, the training tasks on Edge nodes are not supported yet with flexible dynamic migration between Edge and Cloud. This paper proposes a Cloud-Edge AI microservice architecture, based on Osmotic Computing principles. Notable features include: (i) containerized architecture enabling training and inference on the Edge, Cloud, or both, exploiting computational resources opportunistically to reach the best prediction accuracy; and (ii) microservice encapsulation of each architectural module, allowing a direct mapping with Commercial-Off-The-Shelf (COTS) components. Grounding on the proposed architecture: (i) a prototype has been realized with commodity hardware leveraging open-source software technologies; and (ii) it has been then used in a small-scale intelligent manufacturing case study, carrying out experiments. The obtained results validate the feasibility and key benefits of the approach.

## 1. Introduction

Cyber-Physical Systems (CPSs) integrate computational, physical sensing/actuation, environmental, and human components to implement, control, and automate complex processes in the real world [1]. Integration has grown deeper and wider in the latest decade, thanks to technological advances in the so-called *Internet of Things* (IoT): the adoption of sensing, instrumentation, and computing capabilities in micro- and nano-devices embedded in objects dipped in everyday environments and communicating through low-power, lossy networks connected to the Internet. This evolution has lead to the ability to capture unprecedented amounts of data from natural and human-made settings and processes.

The concurrent evolution of Artificial Intelligence (AI) methods and technologies makes it possible to simulate increasingly accurate rational behavior in machines by learning from data, outperforming expert human professionals in a growing number of tasks [2]. AI techniques like *deep learning* [3] exploit ever larger information corpuses to train Machine Learning (ML) models and provide inference (i.e., prediction) capabilities on measurements with increasing accuracy. While early IoT-based AI solutions uploaded all data to Cloud computing infrastructures for model training and inference, the increasing availability of capable processing devices on boundary nodes of local networks has led to a large adoption of so-called *Edge Computing* [4] for that. Its main goal is to push computation and communication resources from the Cloud toward the edge of networks, to supply services and perform rapid computations, avoiding unnecessary communication latency and enabling faster proxy responses for end users. Edge Computing has quickly grown to a worldwide market revenue of USD 139 billions in 2019, with a forecast of over USD 250 B for 2024 (source: Statista.com (accessed on 11 January 2022)). Following this vision, the new *Edge Intelligence* (EI) paradigm [5] promotes the confluence of Edge Computing and AI. EI is currently implemented as the possibility to run AI models on Edge devices, while the more computationally expensive task of training AI models is still carried out in powerful Cloud data centers. Collaborative Cloud-Edge Intelligence has been identified as the next evolutionary step [6], where model training and prediction tasks can be performed either in Edge or in Cloud nodes, depending on application requirements. Nevertheless, practical solutions and architectures are not available yet, due to the complexity of managing resources and services dynamically across the Edge and Cloud layers. The *Osmotic Computing* (OC) [7] is a recent approach aiming at distributed and federated environments driven by increased capabilities in Edge Computing. In OC, dynamic orchestration enables the automatic deployment and elastic migration of microservices from Edge to Cloud infrastructure nodes and vice versa, in order to optimize availability and performance with respect to variable workloads, network topology, and mobility of devices and resources.

This paper presents a novel Cloud-Edge AI microservice architecture for IoT-oriented CPSs based on the OC paradigm. It supports gathering data streams from local cyber-physical devices, preprocessing them and performing AI model training and inference in ML classification and regression problems. Most notably, the same AI microservices—encapsulated in *containers*—can be deployed either to Edge nodes or to Cloud infrastructure *opportunistically*, i.e., exploiting resources available in the neighborhood of the task to be completed and in the current time frame [8]. Edge AI can grant lower prediction latency and turnaround time, in addition to inherently higher data privacy; conversely, Cloud AI can maximize model accuracy and provide further analytics for the end users. The proposed platform also supports hybrid Cloud-Edge AI solutions.

The main contributions of the proposal are:Containerized AI service architecture enabling training and inference to be performed on the Edge, Cloud, or a combination of the two, exploiting available computational resources opportunistically with different trade-offs between computational/storage requirements and prediction accuracy.Microservice encapsulation of each architecture module with an exact characterization of roles, responsibilities, and interactions, allowing a direct mapping with Commercial-Off-The-Shelf (COTS) components, in order to increase feasibility as well as to reduce development costs and time to market.A fully functional platform prototype implemented on commodity hardware by integrating off-the-shelf open-source software technologies and tools.A case study on Cloud-Edge AI in an intelligent manufacturing scenario, with an experimental campaign to validate key value propositions of the approach.

The remainder of the paper is organized as follows. Section 2 discusses on related work, while the proposal is described in Section 3, focusing on the overall architecture, its individual components, and selected available open-source tools to implement it. Case study and experiments are presented in Section 4, before the conclusion.

## 2. Related Work

The authors in [6] extended the EI definition as a paradigm that fully exploits the available data and resources across the hierarchy of end devices, Edge nodes, and Cloud data centers to optimize the overall performance of training an AI model and inferencing. The AI models can work in a Cloud–Edge device coordination, according to a six-level classification of EI architectures. Moving from “Cloud Intelligence” to “All On-Device”, the amount and path length of data offloading decrease, as well as the transmission latency and bandwidth cost, while data privacy increases. As a consequence, Cloud and Edge resources should be exploited opportunistically when designing complex IoT-oriented computing systems.

The majority of the most recent Cloud-Edge AI platforms belong to the “In-Edge co-inference” level of the above classification: model training occurs only in the Cloud, then inferencing (prediction) is carried out to the Edge. The *GEM-Analytics* platform for energy management [9] is a relevant specimen of this category: models are Cloud-trained and validated, and then periodically sent to Edge nodes, where they are used for day-to-day operations in power plants.

The Osmotic Computing is one of the most actively studied approaches to overcome the difficulty to execute AI tasks coordinating Edge and Cloud layers. An in-depth analysis of current perspectives can be found in [10], presenting main issues and challenges in developing and deploying AI-based applications in an OC environment.

Several works have adopted the OC paradigm in a wide range of scenarios. The trust management framework for Pervasive Online Social Networks (POSNs) in [11] exploits OC for an efficient computational offloading among the many users of a POSN. An OC architecture is also proposed in [12] for a smart classroom where deep learning models are tested to recognize entities and chalkboard handwriting, and to control IoT devices. Adopted OC features, however, are quite rudimentary, as microservices are not containerized and a dynamic orchestration is missing. The *Apollon* OC platform proposed in [13] for pollution monitoring, enables opportunistic filtering and integration of data coming from heterogeneous mobile and IoT devices deployed in urban environments. Analogously, the *RAPTOR* [14] osmotic platform allows the creation, deployment, and integration of flexible data analysis applications, through the orchestration of microservices based on the *R* open-source statistical software.

The latest works extend the core OC properties with more advanced capabilities. The reference architecture in [15], named *Osmosis*, focuses on microservices deployment across Cloud, Edge, and IoT environments. Design principles of an osmotic smart orchestrator are investigated, capable of migrating *MicroELements* (MELs) composed of microservices along with *microdata* they work on. The above architecture is exploited in [16] to define a distributed healthcare system. A Body Area Network (BAN) case study highlights system potentialities. In [17] further architecture and orchestration mechanisms are proposed to implement a Message-Oriented Middleware (MOM) for IoT environments based on OC principles. The framework *En-OsCo* [18] aims towards an energy-aware management of resources. It adopts an extended Kalman filter to monitor Edge data centers and hyper-heuristics for an optimal dispatch of services on incoming workloads. The Mobile Augmented Reality Network (MARN) architecture in [19] exploits OC for migrating and effectively scheduling various services across multiple servers. Key requirements of low latency, robustness, and tolerance are monitored, as they are essential to support distributed mobile augmented and virtual reality applications.

Table 1 summarizes relevant features of the above frameworks with respect to the approach proposed here. In particular, our solution is the only one using both osmotic orchestration of containerized microservices and Cloud-Edge Intelligence, allowing data mining with predictive ML models trained and executed on Edge and on Cloud.

## 3. Osmotic Cloud-Edge Architecture

The proposed approach is based on the architecture depicted in Figure 1. Microservices—marked with little green cubes—are encapsulated in containers and opportunistically deployed to devices. A container includes only well-defined components of the operating system (OS), middleware, and application-level software, as required to run a specific (micro)service. Exploiting OS-level virtualization, containers lead to significantly lower distribution overhead and higher density of instances per device than using a hypervisor. Therefore, the new container-based approaches allow the implementation of lightweight services on resource-constrained programmable Edge devices such as gateways, network switches and routers; in the same way, they increase the performance of the dynamic management of microservices within Cloud data centers.

In the reference architecture, the orchestration and provisioning of different containers on the available devices follow the Osmotic Computing principles. Strategies for service orchestration take into account the requirements of both the infrastructure (such as load balancing, reliability, and availability) and applications (such as sensing and actuation capabilities, context awareness, topological proximity, and Quality of Service—QoS—parameters), both changing over time. For this reason, it is necessary to manage a bi-directional flow of microservices between Cloud and Edge. Due to the high heterogeneity of physical resources, the provisioning of containers must adapt the virtual environment to the destination hardware equipment. Furthermore, the migration of services in the Cloud-Edge system needs dynamic and efficient administration of virtual network resources to avoid application failures or QoS degradation. To address these issues, the management of data and applications (*data plane*) is separated from the control of network and security services (*control plane*). The OC paradigm supports this approach, providing a flexible infrastructure for the automatic and secure provisioning of microservices.

As shown in Figure 1, the proposed architecture spans two main infrastructural layers: Cloud and Edge. In the Cloud, data centers host different types of services composed according to high-level application requirements. The Edge level identifies the computing environment at the border of the local network, between local IoT devices and the Internet. It includes data acquisition points and gateway nodes, capable of performing computations on data produced by end devices. The latter gather raw data with a frequency that depends on a variety of factors, including:Rate of change of the observed phenomenon;Environment and context constraints;Ability of the device itself to collect or record data;Operational system requirements to be met.

Due to computational resource constraints, minimal or no data preprocessing is possible on Edge components. They basically act on the raw data collected in the environment, preparing for mining workflows:Decrypting incoming data streams and encrypting outgoing ones, for security;Transcoding data streams between different formats;Combining multiple data streams from groups of devices;Preprocessing and filtering streams to eliminate spurious data, noise, and artifacts;Summarizing raw data to reduce volumes with minimal information loss.

In conventional architectures, the most advanced and computationally complex tasks are reserved to the Cloud infrastructure, including all the steps to train and use ML models:Advanced preprocessing of input data streams, including e.g., function transforms to frequency domain representations;Feature extraction and selection for data dimensionality reduction;Model training from features;Prediction using the trained model.

On the contrary, in the architecture proposed here, Edge nodes can perform either prediction by means of pre-trained models or even the full feature extraction—training—prediction workflow. Due to the different requirements and capabilities of Cloud and Edge systems, it makes sense to devise a heterogeneous architecture where different types of resources are distributed on the two layers. In particular, the Edge level typically includes components with significantly more restricted processing and memory. As a consequence, model training and prediction should be deployed at the Edge when prediction accuracy has lower priority than other requirements, such as minimizing response latency or preserving data locality due to privacy concerns; in those cases, sub-optimal accuracy is an acceptable trade-off.

For these reasons, it is essential to characterize the way composite microservices must be automatically adapted to deployment sites, considering the location and the context of distribution, since containers performance is closely related to the capabilities of the physical host. Additionally, the orchestrator binds at runtime each microservice to its reference location, based on constraints identified by the specific application and by the infrastructure provider. Therefore, as shown in Figure 1, a dynamic service orchestration, based on a feedback loop to detect changes in infrastructure performance and QoS metrics, is achieved using the following logical components:One or more *Edge nodes* programmed to acquire raw data and to process them locally using machine learning algorithms for classification or regression tasks.One or more *Cloud nodes* able to receive aggregate data from the Edge nodes and perform classification/regression tasks by operating on a larger and more articulated data set, while also being able to act as backup hosts for Edge microservices in the case of unavailability of Edge nodes.A *Data Stream Management System* (DSMS) capable of conveying data coming between the Edge of the network towards the Cloud components, while also providing support for data storage operations.An *Orchestrator*, following the OC paradigm to manage different containers implementing the required functional blocks as microservices.

Individual components are detailed in the next subsection, while technological choices for their reference implementation and integration are explained in the subsequent one.

### 3.1. Microservices

The proposed framework leverages a collection of lightweight services, which are loosely coupled and enable granular scalability and flexible composition patterns to cater to both requirements and constraints of applications. In more detail, the following services are proposed.

**Local Storage:** Stores locally and temporarily the data gathered from IoT field devices. Due to latency and bandwidth optimization, centralized and shared data storage should be avoided. Each data processing microservice requires the data to be located as close as possible. This service provides simple access mechanisms like RESTful (REpresentational State Transfer) APIs (Application Programming Interfaces) or event-driven interaction.

**Data Processing:** Performs preprocessing for subsequent ML model training tasks. For complex and high-volume CPSs, the overall task is distributed among Edge nodes to spread the computational load and exploit data locality. In fact, data from the local storage service are accessed directly, thereby reducing bandwidth consumption and latency, while also mitigating common issues of typical Edge devices, such as power and connectivity outages.

**Data Stream Management System** (DSMS): Acts as Message Broker (MB) for the platform, forwarding data and event streams from Edge to Cloud and vice versa. It adopts the *publish/subscribe* pattern, enabling efficient event-driven asynchronous communication. This paradigm is particularly well suited for microservice architectures [17]. Each Edge node can send messages to unique *topics* marking different information types. Each topic has zero to many consumers subscribing to it, and refers either to raw or preprocessed data streams, or to events and control messages. The DSMS allows also the discovery of available topics, published by data producers at the Edge. Furthermore, it is an event stream processor, by combining and possibly converting input data from multiple selected topics to produce an output flow which is subsequently processed by the Cloud Intelligence modules to train a global model. It is also crucial for the DSMS to be interoperable with the most widespread IoT communication protocols, such as *MQTT* (Message Queuing Telemetry Transport) [20] or *CoAP* (Constrained Application Protocol) [21], as many kinds of commercial IoT devices cannot be upgraded to support new protocols due to proprietary firmwares as well as computational resource and deployment capability limitations.

**Data Producer:** Sends data from an Edge node to other Edge or Cloud nodes. The message broker supports the connection.

**Data Consumer:** Receives data sent by a data producer. It is typically deployed on Cloud nodes to get preprocessed information from Edge nodes, in order to be mined.

**Edge Intelligence:** Executes algorithms on Edge devices for ML problems like classification and regression. This microservice is also able to provide model training and validation, based on the data provided by the local storage. The following benefits ensue: (i) privacy and security, as the transfer of sensitive data across the Internet can be avoided; (ii) low latency, as the local model can be trained without waiting data upload to the Cloud and prediction or model download from it; (iii) scalability, as distributed learning is able to manage high volumes of data produced in real IoT-based CPSs.

**Cloud Intelligence:** The Cloud counterpart of the Edge Intelligence service. It runs ML algorithms on data streams produced at the Edge. For example, the Cloud node can train a classification or regression model on streamed sensor data, collected from multiple data producer instances. This approach enables a feedback control loop to update the model and improve its quality progressively; a less accurate model can be trained and used on Edge devices, while a more accurate one is trained in the Cloud by collecting larger amounts of data and is then transferred to the Edge. This loop can be repeated periodically when new data are collected.

**Data Analytics:** Carries out further business intelligence analytics on data gathered at the Cloud layer. In particular, it provides functionalities and tools to support a presentation layer, e.g., a dashboard where aggregated statistics as well as predictions and performance of trained models can be reported.

**Orchestrator:** Manages the aforementioned microservices by means of a container-based approach. In particular, it schedules the migration of services from Edge to Cloud and vice versa, based on real-time resource conditions and availability. For example, the orchestrator can reassign containers in case of network infrastructure changes, high service demand, or Edge node failures.

### 3.2. Technologies

One of the key goals of the proposed platform architecture is to enable the realization and integration of autonomous microservices by exploiting Commercial-Off-The-Shelf (COTS) software components to implement both system functionalities and basic application modules, allowing for lower platform development time and effort.

Table 2 illustrates the mapping of each architecture component (Figure 1) with the corresponding selected technology. An extensive and in-depth market research has been carried out to select tools suitable for each microservice, evaluating functionalities, technological characteristics, costs, licenses, and hardware and software requirements. COTS components with the following features have been preferred:Open source software license with an active developer community;Proven track record of reliability, security, and performance;Full compatibility with container technologies;Interoperability with widespread IoT technologies and protocols;Support for multiple hardware architectures;Support for innovative functional and architectural methodologies of software engineering.

The following technologies were selected to implement and integrate the proposed platform:

**balenaOS:** (https://www.balena.io/os (accessed on 11 January 2022)) A lightweight operating system based on the *Yocto* project (https://www.yoctoproject.org (accessed on 11 January 2022)) for Linux distribution customization. balenaOS is tailored to run application containers on single-board computers and embedded devices. The OS provides robust networking functionalities as well as virtualization and provisioning support. For container management balenaOS includes *balenaEngine*, a *Docker* (https://www.docker.com (accessed on 11 January 2022))-compatible daemon optimized for application service images, and containers and volumes deployed on resource-constrained devices. With respect to other existing container technologies, this tool overcomes common virtualization problems related to embedded scenarios such as resource overhead and lack of hardware support, as balenaOS is available for several device types and different CPU architectures.

**openBalena:** (https://www.balena.io/open (accessed on 11 January 2022)) A balenaOS-based provisioning and orchestration platform to deploy and manage containers on fleets of devices. It is exploited to configure application containers, push updates, share network parameters and distribute container images on each device according to multiple strategies.

**Apache Kafka:** (https://kafka.apache.org (accessed on 11 January 2022)) A distributed event streaming platform for communication among several devices and applications, characterized by horizontal scalability, high throughput, low latency, and interoperability with existing IoT communication protocols through an ecosystem of plug-ins and connectors. Kafka has been adopted as DSMS for sending/receiving streams of event data collected by the container applications. Messages can also contain event feedback forwarded to Edge nodes and the outputs of the ML algorithms exchanged between Cloud and Edge modules.

**Kafka Producer/Consumer API:** (https://kafka.apache.org/documentation/#api (accessed on 11 January 2022)) Each microservice can produce (i.e., send) or consume (i.e., receive) data through the Kafka API. The producer API allows containers to send data to other services subscribed to the same topics, whereas the consumer API can be exploited to retrieve information marked with specific topics in the Kafka platform. Both APIs are available for several programming languages; Python was chosen as it facilitated integration with other Python-based platform components, like the ML APIs and the custom scripts developed for the Data Processing microservice.

**Redis:** (https://redis.io (accessed on 11 January 2022)) In-memory data store used to collect information coming from sensors and field devices according to a key-value data model. Several features make it appropriate for Edge computing scenarios: (i) low CPU and memory requirements; (ii) lightweight data structures particularly appropriate for time-series data; (iii) simple but versatile data model, useful to store information produced by heterogeneous devices; and (iv) append-only storage options optimized for flash memories usually endowing IoT devices.

**TensorFlow:** (https://www.tensorflow.org (accessed on 11 January 2022)) An open source machine learning library, exploited to process the collected data on both Edge devices and Cloud nodes. The *Keras* (https://keras.io (accessed on 11 January 2022)) high-level API has been used to define and train classification and regression models based on deep neural networks, as well as to make predictions on data.

**Streamlit:** (https://www.streamlit.io (accessed on 11 January 2022)) Python-based library used to create interactive Web applications able to: (i) plot sensor data, also highlighting basic statistics and patterns; (ii) support exploratory data analysis; and (iii) visualize performance results of ML predictive models.

## 4. Case Study: Intelligent Manufacturing

Intelligent manufacturing is a challenging cyber-physical system case study. The increasing adoption of IoT and AI technologies, spurred by policy initiatives like *Industry 4.0* [22], is transforming manufacturing with significant organizational, economic, and societal impacts. Plants are shifting from collections of big monolithic machines and robots to large networks of smaller, individually controllable sensing and actuation components, with multiple continuous data streams feeding various distributed decision points, either autonomous or under human supervision.

The reference scenario considered here concerns impurity prediction on iron concentrate in the mining industry: *An iron extraction plant operates in an industrial area. Process variables and surrounding air flow must be continuously monitored, carrying out an autonomous intelligent manufacturing task to maximize mineral quality.* Mining activities face a constant decrease in ore concentration. Several processing techniques are currently used to increase the recovery of ore from raw materials and represent fundamental operations of modern separation processes used in the mining industry. *Flotation* is one of the most widely used techniques, allowing the separation of gangue from ore. However, since the impurity (silica) in iron ore is commonly measured every hour, being able to predict the amount of impurity could constantly support the activity of engineers and technicians, by providing useful information in advance to promptly improve the extraction process.

A monitoring platform prototype was developed, based on the proposed architecture. Sensors and actuators embedded in the extraction plant were simulated by means of the reference dataset in [23]. Inspection features include starch and amina (reagents) flow, ore pulp flow, ore pulp pH, and ore pulp density, which are the most important variables for the final mineral quality. Further data include level and air flow inside the flotation columns, i.e., cylinders where mineral slurry and air flows are introduced from above and from below, respectively, in order to induce mixing. The proposed architecture is able to aggregate these data opportunistically and predict the amount of silica in the extraction process by means of Cloud-Edge Intelligence algorithms.

### 4.1. Prototype

Following the architecture described in Section 3, a prototypical testbed was developed to prove the feasibility of the proposal and to evaluate its performance and capabilities. The main components are depicted in Figure 2. It is an industrial IoT-based CPS environment consisting of two Edge nodes (i.e., two independent plant sectors), an Orchestrator and Message Broker (MB in the following), and a Cloud node. Edge devices and the MB are connected through an IEEE 802.11 wireless local network, and the MB is the only module communicating with the remote Cloud node through an Internet connection.

As depicted in Figure 3, different single-board computers were employed on the Edge side. With reference to Figure 2, each microservice was deployed on a different board: this shows how, in the proposed framework, logically related microservices can be distributed across multiple hardware devices instead of being confined in a single node. The system scales horizontally according to available devices within the target environment. For example, devices with limited computational resources may be used as Storage modules, i.e., to collect data from physical sensors, whereas boards with higher-performance CPUs can be exploited for more compute-intensive tasks. High modularity and scalability are the main benefits of the adopted OC approach.

A Raspberry Pi 4 Model B (https://www.raspberrypi.org/products/raspberry-pi-4-model-b (accessed on 11 January 2022)) (RPi4) was used as the MB, running an instance of Apache Kafka, a MQTT to Kafka connector, and the Orchestration service in different balenaOS-based containers. It is equipped with a quad-core 1.5 GHz ARM64 CPU, 4 GB of RAM, and 32 GB of Secure Digital (SD) storage memory. As per the publish/subscribe pattern, each message published by any node to a topic is received by all subscribers for the topic. The following topics have been defined: control, used to transmit messages regarding the (dis)connection of Sensor nodes, or data availability in a Storage module; data, used to share messages related to data processing and results of the inference algorithms.

Each Edge node from Figure 2 is composed of two devices running different containerized modules on balenaOS. The first Edge node ($E_{1}$) includes a Raspberry Pi 3 Model B+ (https://www.raspberrypi.org/products/raspberry-pi-3-model-b-plus (accessed on 11 January 2022)) (RPi3+) to perform Edge Intelligence tasks ($E_{1a}$), and a local Storage module running on a Raspberry Pi 1 Model B (RPi) ($E_{1b}$). The RPi3+ is equipped with a quad core 1.4 GHz ARM64 CPU, 1 GB RAM, and 32 GB storage memory, whereas the RPi has a single-core ARM11 CPU at 700 MHz, 512 MB RAM, and 8 GB SD storage memory. Similarly, the second Edge node ($E_{2}$) was configured using a Raspberry Pi 3 Model B (https://www.raspberrypi.org/products/raspberry-pi-3-model-b (accessed on 11 January 2022)) (RPi3) equipped with a slightly slower quad core 1.2 GHz ARM64 CPU, 1 GB RAM, and 32 GB SD storage memory ($E_{2a}$), which runs the Edge Intelligence service, and an RPi acting as a second Storage device. Additionally, two RPi devices ($E_{1c}$ and $E_{2c}$) with similar specifications were used to simulate a sensor network, sending raw data to the Storage devices via dedicated MQTT topics.

The Cloud node containers were deployed to a remote Microsoft Azure D32as v5 virtual machine, configured with an Intel Xeon CPU E5-2673, 32 GB of RAM and 128 GB of storage. The Internet connection between the local network (including the Message Broker) and the Cloud is an asymmetric Fiber To The Cabinet (FTTC) small office link with downstream and upstream nominal bandwidth of 100 Mbps and 30 Mbps, respectively.

Different message formats have been used for communication. Simulated Sensor devices communicate with the MB to discover available Storage modules, then they start transmitting data serialized in the *Apache Arrow* (https://arrow.apache.org (accessed on 11 January 2022)) format, a language-independent standard proposed for general-purpose serialization and data transfer. Since it is based on a column-oriented layout, the Arrow format is particularly suitable for tasks requiring fast data processing and information sharing between Storage devices. On the other hand, all messages transmitted through the MB via the control and data topics are serialized in *JSON* (JavaScript Object Notation) (https://www.json.org (accessed on 11 January 2022)) format. In particular, control messages have the following attributes, summarized in Table 3:*id*: Unique message identifier;*type*: Indicates the kind of control message. Acceptable values are:
–*storage_connected* (SC): A new Storage module is available on the network;–*storage_disconnected* (SD): A Storage module is currently unreachable or down;–*sensor_data* (SDT): A Sensor measurement is available on a storage module for running a prediction algorithm;–*dataset* (DS): A Sensor dataset, including several sensor measurements, is available on a Storage module for training or updating the ML models;–*query* (QR): Used to list available Storage modules and related data;–*response* (RS): Indicates a response to a query message.*host*: Contains the reference module IP address;*data_key*: Unique identifier used to retrieve data from a specific Redis datastore;*query_type*: Used to retrieve information about a single measurement (*sensor_data*), a subset of data (*dataset*) or the whole collection (*storage*);*query_id*: Message id of the query originating current response;*storage_id*: Unique identifier of the Storage module containing the data.

Table 4 summarizes the attributes of data messages:*type*: Indicates the kind of data notification:
–*input*: Contains data samples for which an inference task is requested;–*output*: Contains results of a prediction task;–*model*: Returns information about the performance of the trained ML models.*id*: Identifies the processed sensor data (in case of input and output messages) or the Cloud/Edge Intelligence module providing the prediction model;*data*: An array of raw information;*module_id*: Identifies the Intelligence node running the predictive algorithm;*result*: Output of the prediction task;*time*: Prediction time in milliseconds;*r2*: Coefficient of determination ($R^{2}$), used as a performance metric for a regression model;*mse*: Mean squared error, i.e., the average squared difference between predicted and real values;*download_time*, *training_time*, and *evaluation_time*: Time spent by the Intelligence node to retrieve the whole dataset, train the model, and evaluate performance, respectively.

An ore purity monitoring and prediction process was developed with a regression model through the above prototype. Basically, it entails the sequence of interactions reported in the UML (*Unified Modeling Language*) diagram of Figure 4 and described in what follows:1.When a Storage module is available on the network, it sends a *storage_connect* control message to notify all Sensor modules subscribed to the control topic (blue messages in Figure 4). As an alternative, each Sensor module can explicitly perform a query to retrieve all the available Storage devices.2.The Sensor module collects data during its observation period and sends it to Storage devices through a dedicated MQTT topic. The Apache Arrow data format is used for message serialization (red color in Figure 4).3.A *dataset* notification is sent to advertise the availability of new data. Datasets can be used to train or update prediction algorithms on active Cloud/Edge modules, but also to plot information on a remote dashboard. Intelligence modules can autonomously query the MB to obtain information about available datasets.4.Data are retrieved from one or more Storage devices and used to train a regression model. Performance results are then exposed through a *model* message on the data topic (drawn in orange in Figure 4).5.Subsequently collected sensor data represent the *input* of the prediction model and are forwarded through the MB to the subscribed Intelligence nodes. Results of the regression process are finally returned through an *output* notification.

Final prototype specification concerns the ML model trained in the Intelligence nodes for the case study: It is a *multi-layer perceptron* regressor [3] with 5 hidden layers and 200 neurons per layer, with a *Rectified Linear Unit* (ReLU) activation function. The network is trained for 10 epochs using the Keras implementation of the *Adam* [24] optimizer, with default parameters and mean squared error loss function. This model was selected because it generally provides satisfactory prediction performance, while being sufficiently lightweight to run in a timely manner on resource-constrained devices.

### 4.2. Experiments

An experimental campaign was carried out to assess the prototype performance. The whole set of data [23] adopted to simulate the intelligent manufacturing use case contains N=737453 samples, collected in a 7-month time span, for a 160 MB total size. The deployed architecture is the one described in Section 4.1, where each logical node is composed of two devices: Edge Intelligence and Storage. In order to test the dependency of performance on dataset size and to simulate deployments on a larger scale than what was allowed by available physical devices, four scenarios were configured, with data dimension set to $N,\frac{N}{2},\frac{N}{4},\frac{N}{8}$, respectively, and samples extracted randomly. In each configuration, a validation set was obtained by holding out 1/7 of the dataset, and the remaining 6/7 were used to train the predictive models, with 3/7 for the two simulated Sensor devices, $E_{1c}$ and $E_{2c}$. With regards to all experimental results, each reported value is the average of five cold runs.

**Data gathering.** The first test simulated data upload by Sensor devices to Storage. To reduce network load and memory usage, due to Redis being an in-memory store, data were first compressed using the *zlib* format [25]. Basically, compression increases CPU usage on both Sensor and Edge devices, while a range of network bandwidth and RAM to CPU usage trade-offs can be achieved by tuning the compression parameters, or by replacing the format altogether, so as to meet scenario requirements. Data upload was carried out according to steps 1–3 of the sequence in Section 4.1. Elapsed times have been reported in Figure 5 for each above scenario. The results show linear dependence of import time on the number of samples.

**Model training and validation.** The second test involved measuring training times and the related network load. As explained in steps 4–5 of the sequence in Section 4.1, each Intelligence module needs to fetch data from the Storage before training. On training completion, it performs predictions on the validation set. Table 5 reports on network loads for this phase, with device labels referring to Figure 3. Predictably, the busiest modules from the network point of view are the Storage ones, which both receive information from Sensors and upload them to requesting nodes. This seemingly inefficient approach, however, decouples data production of field devices from data consumption of Edge Intelligence modules, which is useful as they may have significantly different velocity and/or variability.

In order to compare the performance of the proposal against a centralized Cloud solution, the same Intelligence container used in the Edge nodes was deployed on the Cloud via the OC Orchestrator on the MB. The Cloud node first downloads the full dataset by querying all the Edge storage devices, then proceeds to train a predictive model. Turnaround times and model validation results are reported in Table 6. Considering that Edge nodes use half of the samples to train their models, it can be noticed how Storage data retrieval takes a comparable amount of time for both Cloud and Edge nodes, while training time is significantly shorter on the Cloud (almost an order of magnitude), as expected due to its more powerful hardware. While this may be mitigated by deploying Edge Intelligence services on more capable devices, it is not a crucial issue, as training on the whole dataset only happens once, or at worst periodically, depending on the use case. Actually, in real applications it is advisable to train models periodically and incrementally, using small-size datasets. As explained in Section 3, the prototype also allows for Cloud-Edge cooperation in model training: while Edge components can train models “on-the-fly” on smaller datasets, retaining their independence from the Cloud albeit with sub-optimal accuracy, a Cloud node may aggregate more data coming from multiple local networks in order to train better models, finally feeding them back to the Edge.

It is also important to point out that the prototype represents a rather optimistic setting for the Cloud node, where it is both available and mostly idle w.r.t. hardware and network resources. In a realistic industrial scenario, where the premise’s Internet connection towards the Cloud is shared by a large number of devices, the uplink may be temporarily unavailable or may be saturated by sensor data and inference requests, making it a potential bottleneck. Similarly, the full processing resources of the Cloud node(s) will be shared across highly heterogeneous workloads, and company budget pressures will induce IT officers to seek a relatively high utilization baseline [26]. In such settings, Edge Intelligence capabilities can improve both the availability and the timeliness of predictions, by scaling the workload across multiple relatively capable boards located near data generators.

**Dataset size dependency.**Table 7 reports on training times and validation metrics ($R^{2}$ and MSE) for different dataset sizes on the Edge Intelligence node $E_{1a}$. The results indicate there is an acceptable trade-off between model accuracy and training time in Edge Intelligence applications, in agreement with existing evidence suggesting how fractional datasets do not induce a large degradation in prediction accuracy if their distribution is representative of the whole dataset [27]. This outcome supports the aforementioned claims about Cloud-Edge Intelligence cooperation. Additionally, it is important to note that the prototype was set up with separate devices for data storage and model training for experimentation purposes. In real scenarios, the OC Orchestrator may eliminate download latency by deploying Edge Intelligence and Storage microservices on the same component, provided it has enough resources; this would both eliminate download time and reduce the overall network load.

**Prediction performance.** Further experiments have been carried out to evaluate both computational and network performance in prediction tasks. Sensor device $E_{1c}$ was configured to send 10 *input* messages on the *data* topic. Subscribed Intelligence nodes compute predictions and return *output* response messages. Table 8 shows data exchange for this step is minimal, as expected. Sending individual data samples through the MB incurs in some network bandwidth overhead, though this can be mitigated by publishing multiple samples together, if possible. The last experiment assessed prediction time and latencies both at the Edge and on the Cloud. Outcomes are reported in Table 9:*Inference time*: The time elapsed in predicting the regression value for a sample locally, as measured by the Intelligence module.*Communication latency*: The time required for sending and receiving messages between the different components of the architecture in the prediction phase. As Table 9 shows, it is made of four components: (i) from Sensor to Message Broker (S to MB), (ii) from Message Broker to Intelligence (MB to I), (iii) from Intelligence to Message Broker (I to MB), (iv) and from Message Broker to Sensor (MB to S). (In the prototype the last two components simply concern the prediction values, but in general scenarios they could concern set points for appropriate actuators in a control feedback loop, computed on the basis of the ML predictions);*Turnaround time*: The overall time between input sample upload and prediction, evaluated on the Sensor node uploading the samples.

While prediction time is significantly lower on the Cloud, the Edge nodes have notably lower communication latencies. Although the overall turnaround time is lower in the Cloud case, the actual availability of a stable connection is far from certain in a real industrial plant. In fact, whenever bandwidth availability decreases due to other computing and/or control activities deriving from thousands of devices interacting with the Cloud—a condition which is not replicated in our experimental setup— transferring training and prediction to the Edge may improve the overall system performance by alleviating some of the outbound network pressure.

**Osmotic microservice allocation.** In order to assess the capability of the proposed architecture to dynamically adapt to variable workloads and node availability by migrating microservices between the Edge and the Cloud, two alternative scenarios to the baseline behavior described in Section 4.1 were tested:1.A Sensor module looks for an available Storage service, but the Orchestrator is unable to meet the request due to the unavailability of Storage microservices instances and to the lack of a suitable device to host them. The Orchestrator therefore pushes the Storage container to the Cloud, which—once ready—announces itself through a *storage_connect* message. The Orchestrator is now able to notify the Sensor module, which can then upload its data. At some point in time, a new Edge device with the required capabilities to act as a Storage host connects to the network. For load balancing purpose, the Orchestrator hangs the Storage microservice to the new device, which can take over the role of Sensor data collector.2.In case of a shortage of Edge Intelligence nodes (e.g., due to device failure), the Orchestrator pushes the Intelligence microservice to the Cloud, as it can resume its customary learning and inference tasks, albeit with higher network latency. Eventually, a new device connects to the Edge, and the Orchestrator assesses it as being able to host an instance of the Edge Intelligence service. The microservice is therefore linked to the new device, thus offloading the Cloud and restoring normal operation.

The main difference from the baseline essentially consists of the Orchestrator pushing containers to Cloud and Edge nodes, while the rest of the system keeps working as usual. Microservice deployment times and bandwidth usage are reported in Table 10: The *load time* column reports on the time taken to address the container from the Orchestrator to the target node and load it in the Balena engine. Conversely, *startup time* refers to the time it takes to bootstrap the container after it has been loaded. Both load and startup times are higher for Edge nodes, as expected due to being significantly less capable than the Cloud, but still within one order of magnitude. Times for the Intelligence container are much closer, which is likely due to the allotted Edge device being a RPi3+, versus a RPi for the Storage container; this hints at the dominance of container loading time in the Balena engine over network transfer time for slower devices. Bandwidth usage has been measured on the Orchestrator, and it is generally lower for the Edge node; this is due to the size of container images being different between the target devices, as reported in the *image size* column.

Finally, Figure 6 shows dataset upload and download times when the Storage microservice is deployed to the Edge and Cloud nodes, representing Sensor data compression and upload in the first scenario, and dataset download and decompression by Edge Intelligence nodes in the second scenario, respectively. Processing time is mostly noticeable in the upload phase, as data is compressed by an RPi device before uploading them to the Redis data store, while it is negligible when an RPi3 device decompresses the downloaded dataset. Transfer time is significantly higher for the Cloud, which is due to the network connection being asymmetrical, with much higher downlink than uplink bandwidth. In any case, transfer time is consistently higher when the Storage microservice is deployed to the Cloud, as expected, though again in the same order of magnitude. This confirms that the elastic—osmotic—allocation of the Storage microservice between the Edge and the Cloud is a viable solution which can be carried out without significant impact on the overall system performance.

## 5. Conclusions and Future Work

This paper presented a novel Cloud-Edge Intelligence distributed framework, mainly targeted at IoT-based Cyber-Physical Systems. It adopts a microservice architecture and follows the Osmotic Computing paradigm to allow opportunistic resource exploitation by means of dynamic flexible deployment of service modules to various devices at the Edge of the network and/or in the Cloud. Clear encapsulation of logical components with well-defined roles and responsibilities enhanced modularity, further enabling a direct mapping with COTS technologies to increase feasibility and reduce development costs and time to market. In the proposed approach, model training and prediction tasks can be performed in Edge or Cloud nodes, but also through Cloud-Edge collaboration. Less sophisticated models are trained and used at the Edge for early response, while Cloud devices periodically train larger and more accurate models, feeding them back to the Edge. Core claims were supported by experiments carried out on a public domain industrial dataset with a full prototypical platform implementation integrating several open source off-the-shelf tools.

Future work concerns:Exploiting knowledge representation and reasoning in the orchestrator to dynamically discover the best deployment configuration via context-aware *semantic matchmaking* [28] between ontology-based annotations of microservices and devices.Investigation of more advanced IoT-oriented AI algorithms, by enhancing machine learning with semantic technologies [29] and computational argumentation.Further development of the analytics and visualization component, not described in detail in this paper as it is currently at an early stage, even though relevant for the usability of the overall solution.Integration of the platform prototype with real sensors and actuators in a manufacturing setting, followed by new experiments,Additional case studies in challenging IoT-based CPS scenarios, such as (tele)-healthcare, environmental monitoring and urban safety control.

## Figures and Tables

**Figure 1 sensors-22-02166-f001:**
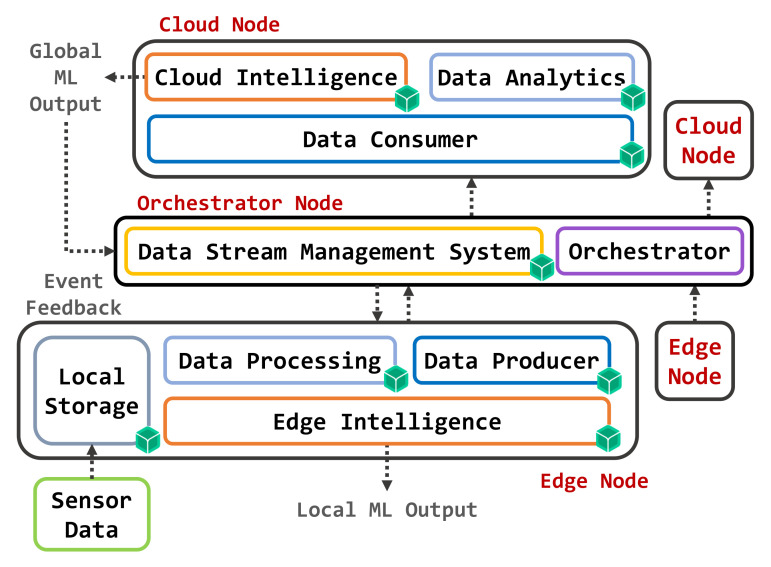
Reference Architecture.

**Figure 2 sensors-22-02166-f002:**
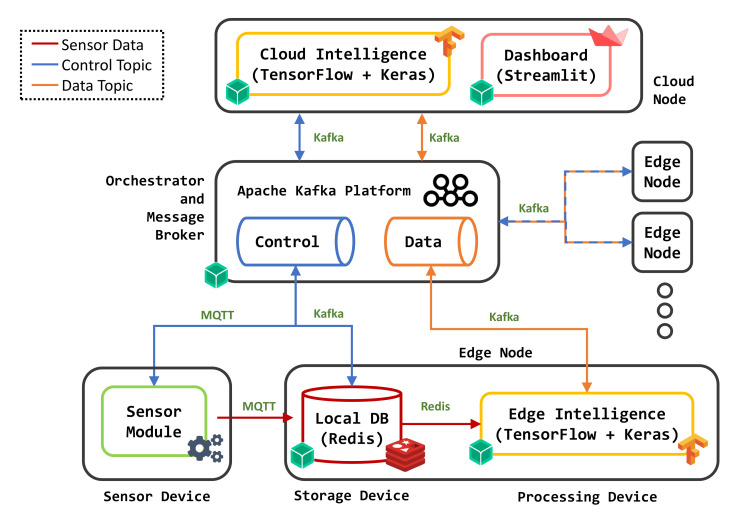
Components of the proposed prototype.

**Figure 3 sensors-22-02166-f003:**
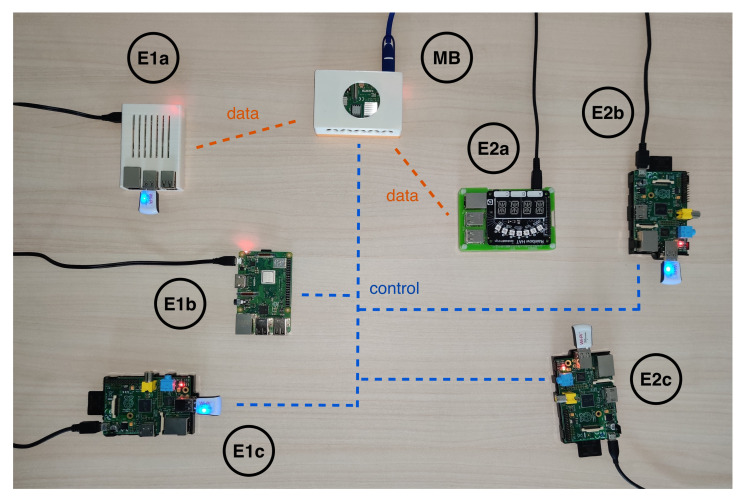
Message Broker/Orchestrator and Edge devices in the platform prototype.

**Figure 4 sensors-22-02166-f004:**
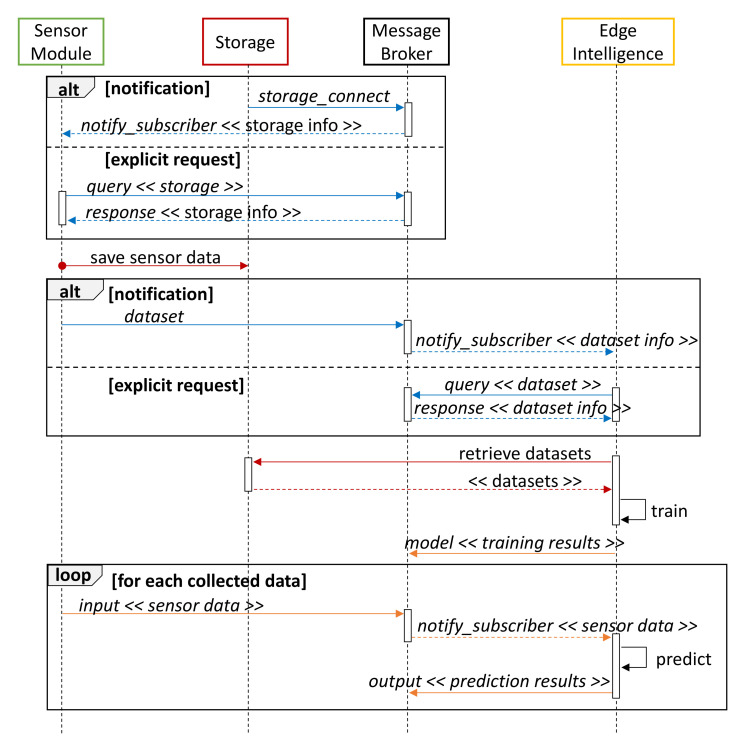
Sequence diagram for Edge-side prediction.

**Figure 5 sensors-22-02166-f005:**
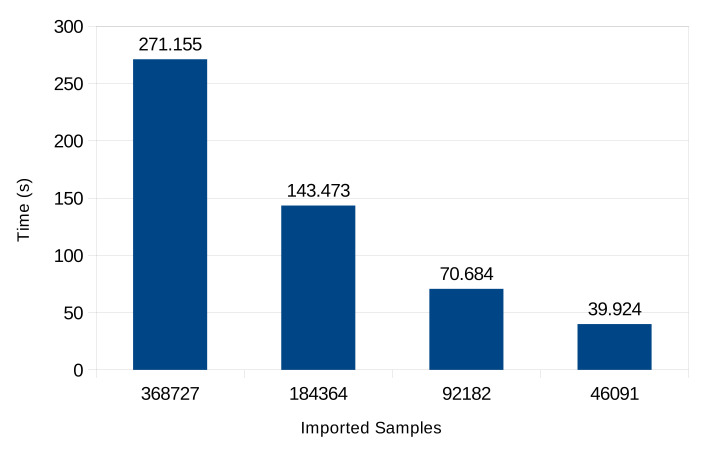
Time required for data import.

**Figure 6 sensors-22-02166-f006:**
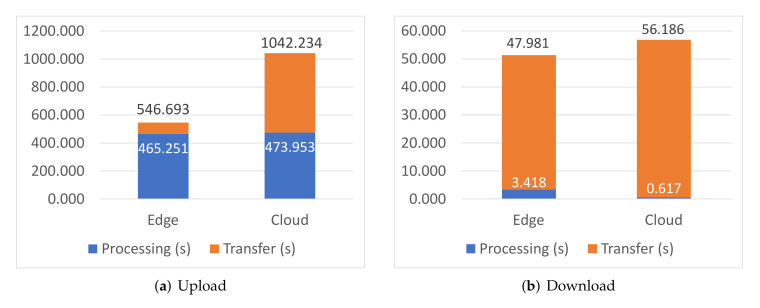
Dataset transfer times.

**Table 1 sensors-22-02166-t001:** Comparative table of frameworks (**✓**: supported, **✗**: not supported).

Reference	Containerized Services	Osmotic Orchestration	AI	Model Training
Pacheco et al. [12]—2018	**✗**	**✗**	**✓**	Pre-Trained
Apollon [13]—2018	**✓**	**✓**	**✓**	On Cloud
Grzelak et al. [14]—2018	**✓**	**✓**	**✓**	**✗**
Carnevale et al. [16]—2019	**✓**	**✓**	**✗**	**✗**
En-OsCo [18]—2019	**✓**	**✓**	**✗**	**✗**
Osmosis [15]—2019	**✓**	**✓**	**✗**	**✗**
Sharma et al. [19]—2020	**✗**	**✓**	**✗**	**✗**
Tovazzi et al. [9]—2020	**✓**	**✗**	**✓**	On Cloud
This work	**✓**	**✓**	**✓**	On Cloud and Edge

**Table 2 sensors-22-02166-t002:** Reference COTS tools.

Service/Module	Technology	Version	License	Release Date
Container technology	balenaOS	2.54.2	Apache 2.0	12 August 2020
Orchestrator	openBalena	3.1.1	GNU Affero GPL 3.0	10 November 2020
Data Stream Management System	Apache Kafka	2.5.0 (with Scala 2.12)	Apache 2.0	15 April 2020
Data Producer	Kafka Producer API	2.0.1-python	Apache 2.0	19 February 2020
Data Consumer	Kafka Consumer API	2.0.1-python	Apache 2.0	19 February 2020
Local Storage	Redis	6.0.9	3-Clause BSD	26 October 2020
Data Processing	Python scripts	3.9.0	PSF & Zero-Clause BSD	5 October 2020
Edge/Cloud Intelligence	TensorFlow Keras API	2.3.1 2.4.3	Apache 2.0 MIT	12 Sepember 2020 25 June 2020
Data Analytics & Visualization	Streamlit	0.72.0	Apache 2.0	2 December 2020

**Table 3 sensors-22-02166-t003:** Attributes of control messages.

Type	Id	Host	Data Key	Query Type	Query ID	Storage ID
SC	**✓**	**✓**				
SD	**✓**					
SDT	**✓**	**✓**	**✓**			
DS	**✓**	**✓**	**✓**			
QR	**✓**			**✓**		
RS	**✓**				**✓**	**✓**

**Table 4 sensors-22-02166-t004:** Attributes of data messages.

Type	Id	Data	Module ID	Result	Time
input	**✓**	**✓**			
output	**✓**		**✓**	**✓**	**✓**
model	**✓**				
	**R** ${}^{2}$	**MSE**	**Download Time**	**Training Time**	**Evaluation Time**
input					
output					
model	**✓**	**✓**	**✓**	**✓**	**✓**

**Table 5 sensors-22-02166-t005:** Training: Network activity.

Node	Device	Label	Download (kB)	Upload (kB)
Broker	broker	MB	4768	7603
	intelligence	$E_{1a}$	33,386	993
Edge 1	storage	$E_{1b}$	35,951	36,043
	sensor	$E_{1c}$	2379	35,259
	intelligence	$E_{2a}$	33,685	1142
Edge 2	storage	$E_{2b}$	35,522	35,858
	sensor	$E_{2c}$	2193	35,088

**Table 6 sensors-22-02166-t006:** Training time and validation results.

Node	R${}^{2}$	MSE	Download Time (s)	Training Time (s)	Validation Time (s)
Cloud	0.983	0.0222	50.788	437.986	3.081
Edge 1	0.972	0.0348	24.603	2086.653	25.415
Edge 2	0.971	0.0337	33.085	2574.005	28.976

**Table 7 sensors-22-02166-t007:** Training: Reduced datasets.

Samples	$R^{2}$	MSE	Download Time (s)	Training Time (s)	Validation Time (s)
737,454	0.983	0.021	42.451	5155.740	32.072
368,727	0.972	0.035	24.603	2086.653	24.608
184,363	0.959	0.055	10.688	1066.539	10.688
92,182	0.931	0.088	5.087	564.569	5.087
46,090	0.894	0.135	2.815	263.953	2.815

**Table 8 sensors-22-02166-t008:** Prediction: Network activity.

Node	Device	Label	Download (kB)	Upload (kB)
broker	broker	MB	42	78
Edge 1	intelligence	$E_{1a}$	26	26
	sensor	$E_{1c}$	36	34
Edge 2	intelligence	$E_{2a}$	26	26

**Table 9 sensors-22-02166-t009:** Prediction: Time and latency.

Node	Inference Time	Communication Latency (ms)				Turnaround Time
	(ms)	S to MB	MB to I	I to MB	MB to S	(ms)
Cloud	31.377	91.510	19.752	42.549	44.970	230.158
Edge 1	230.598	87.259	4.362	19.461	37.773	379.453
Edge 2	301.887	84.812	7.590	22.844	30.555	447.688

**Table 10 sensors-22-02166-t010:** Microservice deployment performance.

Node	Service	Load Time (s)	Startup Time (s)	Broker Bandwidth (MB)		Image Size (MB)
				Download	Upload	
Cloud	Storage	71.163	2.983	4.144	255.726	239.974
	Intelligence	224.451	2.768	12.648	765.300	718.205
Edge	Storage	314.577	23.424	1.713	149.924	137.782
	Intelligence	244.422	3.108	3.424	718.291	694.802

## Data Availability

Code and data derived from the research are under non-disclosure agreement (NDA) with Exprivia S.p.A.

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
