# Peer review of "Osmotic Cloud-Edge Intelligence for IoT-Based Cyber-Physical Systems"

_sensors, 2022, doi:10.3390/s22062166_

Round 1

Reviewer 1 Report

The paper proposes a microservice architecture for IoT applications exploiting Cloud, Edge and AI methods. Its usability is exemplified by an intelligent manufacturing scenario.

The paper is well written and readable, only some typos can be found:

- page 1: "is making it" -> makes it
- page 4: "necessary managing" -> necessary to manage
- page 9: "As pictured in" -> As depicted in

Section 1 well positions the paper by stating background definitions and discussing their combined utilization towards the target research field. The main contributions are well summarized and highlighted at the end of the section. Nevertheless, the abstract should be refined to reflex the same information concerning the contributions.

Section 2 presents the related works, the provided comparison table highlights novelty over them.

Section 3 defines the proposed reference architecture, and discusses the provisioned services and their characteristics. These parts are well described and presented.

Section 4 presents the evaluation. In a prototype implementation, IoT devices are simulated using a historic dataset. Section 4.1 describes the evaluation environment based on the propotype, while Section 4.2 presents the performed measurements. The components (e.g. edge devices) should be placed in a physically more distributed manner. For the first phase of the evaluation, the data gathering is simulated by saving locally the historical data as sensor values. Instead, it would be more realistic to send the sensor data over some IoT protocol (e.g via MQTT).

The evaluation setup is a rather small-scale one, and compares only scenario execution in the cloud vs on two edge devices. This is quite simple as is. One of the contributions stated is to use Osmotic Computing techniques, which is not validated at all. It would be interesting to see, how certain on-the-fly task delegations to the cloud would help out edge devices under various parameter settings. Section 4 should be extended accordingly.

Author Response

We are really grateful to the reviewers for the careful revision they have carried out on our paper. We have done our best to take into account their suggestions and to respond to their requests. Detailed replies are reported in what follows. Modifications to the original manuscript are reported in blue color in the paper.

Reviewer 1

Q 1.1 Some typos can be found:
- page 1: “is making it” -> makes it
- page 4: “necessary managing” -> necessary to manage

- page 9: “As pictured in” -> As depicted in

Response

Thank you very much. We have fixed evidenced typos and we have also fully proofread the manuscript a few times more and striven to improve language wherever needed.

Q1.2 Section 1 well positions the paper by stating background definitions and discussing their combined utilization towards the target research field. The main contributions are well summarized and highlighted at the end of the section. Nevertheless, the abstract should be refined to reflex the same information concerning the contributions.

Response

We are grateful to the reviewer for this suggestion. We have updated the abstract according to the contribution summarized in Section 1.

Q1.3 The components (e.g. edge devices) should be placed in a physically more distributed manner. For the first phase of the evaluation, the data gathering is simulated by saving locally the historical data as sensor values. Instead, it would be more realistic to send the sensor data over some IoT protocol (e.g via MQTT).

Response

Many thanks for this observation. We agree with the remark that incorporating typical IoT protocols like MQTT would make the framework more realistic. We have added IoT protocol support as a further requirement for the Data Stream Management System in Section 3.1, and in Section 3.2 we have highlighted that Kafka can be extended with plug-ins and connectors to support various IoT protocols. We have updated accordingly our reference implementation, by integrating an MQTT connector into Kafka and modifying the overall prototype architecture as displayed in the new version of Figure 2 and described in Section 4.1: the Sensor node now encapsulates one or more physical sensors and communicates with the Message Broker and with the Storage node via MQTT.

Q 1.4 The evaluation setup is a rather small-scale one, and compares only scenario execution in the cloud vs on two edge devices. This is quite simple as is. One of the contributions stated is to use Osmotic Computing techniques, which is not validated at all. It would be interesting to see, how certain on-the-fly task delegations to the cloud would help out edge devices under various parameter settings. Section 4 should be extended accordingly.

Response

We deeply thank the reviewer for this recommendation. Basically, we agree that the number of physical devices involved in the prototype could figure a small-scale experimentation. For this reason, in the original experiments samples import times and training and validation performance have been tested splitting the full dataset in an increasing number of parts (up to 8). This should reflect a scenario where the process feeds 8 Edge Intelligence nodes concurrently. Results are reported in Figure 5 (data import time) and Table 7 (training and validation performance). We have now highlighted more clearly the purpose of those tests in Section 4.2. Anyway, we also agree that on-the-fly task delegation through Osmotic Computing needs some validation. For this reason, we have added a paragraph on osmotic microservice allocation to Section 4.2. There, we report on tests about two different scenarios, concerning Storage and Intelligence microservices pushed to the Cloud when an adequate host is not found in the local network and reallocated whenever a device becomes available again. We have verified the correct behavior of the platform in these cases and we have extracted performance measures about microservice load time, startup time and network bandwidth usage. Results validate the feasibility of the approach, as dynamic service allocation is efficient with respect to the microservice container image size, and business continuity can be granted without losses.

Reviewer 2 Report

Overall the paper is well written and the experimental design is very good. I do have significant concerns about the results presented in Table 9. There is no explanation as to why the brokerage time for the cloud system is so high (1265ms). There are no details about the network latency between the cloud and the broker so it is difficult to tell if it is caused by network delays or is an artifact of the system. The authors should explain the reason for this result as it otherwise looks like an error in the experimental design. Further detail on the network connection between broker and the cloud node would also help.

Author Response

We are really grateful to the reviewers for the careful revision they have carried out on our paper. We have done our best to take into account their suggestions and to respond to their requests. Detailed replies are reported in what follows. Modifications to the original manuscript are reported in blue color in the paper.

Reviewer 2

Q2.1 I do have significant concerns about the results presented in Table 9. There is no explanation as to why the brokerage time for the cloud system is so high (1265ms). There are no details about the network latency between the cloud and the broker so it is dicult to tell if it is caused by network delays or is an artifact of the system. The authors should explain the reason for this result as it otherwise looks like an error in the experimental design. Further detail on the network connection between broker and the cloud node would also help.

Response

The reason for the exceedingly high latency is that the original experiments were executed adopting a mobile phone as gateway to the Cloud: in fact the testbed was built in a home setting due to the inaccessibility of our oces, caused by the COVID-19 pandemics. For a more realistic portrayal of network performance, we have repeated the tests in a small office configuration with 100 Mbps / 20 Mbps asymmetric link to the Internet. Detailed information about the network settings and the adopted Cloud resources have been reported in Section 4.1. New latency results are in the updated Table 9, and they are significantly improved indeed. For greater clarity, we split communication latency in the four segments involved in a typical sample acquisition and prediction response cycle: as expected, Cloud latencies are higher than Edge latencies, but not by much.

This outcome requires to highlight another possibly critical aspect of our tests, which we have discussed in depth in the revised manuscript: our prototype may represent a favorable situation for the Cloud node with respect to real industrial scenarios. Basically, the network link and Cloud virtual machine load were almost completely dedicated to the experiments, whereas in real contexts the premises Internet connection is shared by a large number of devices and processes. In some cases, the stability of Internet connection is not granted. Therefore, the uplink may be temporarily unavailable or may be saturated with various types of data transfers beyond sensor data and inference requests, making it a potential bottleneck. Similarly, the full processing resources of the Cloud node(s) will be shared across highly heterogeneous workloads, and company budget pressures will induce IT officers to seek a relatively high utilization baseline: Cloud virtual machines do not (and should not) typically sit idle. Nevertheless, even in these “ideal” conditions for the Cloud node, results show Edge Intelligence provides the expected benefits.

We are deeply grateful to the reviewer for the insight, as it allows us to decisively improve the manuscript and clarify a relevant element of the experiments.

Round 2

Reviewer 1 Report

The authors addressed all reviewer comments, and the paper has significantly improved. Though higher scale experiments could have been performed, the results are convincing now.

Reviewer 2 Report

The authors have done an admirable job in addressing my concerns. In my opinion the paper is ready for publication.